# Analysis of Production Safety in the Construction Industry of China in 2018

**Xin-Hui Zhou [1], Shui-Long Shen [2],\*, Ye-Shuang Xu [1] and An-Nan Zhou [3]**

[1] State Key Laboratory of Ocean Engineering, Department of Civil Engineering, School of Naval Architecture, Ocean, and Civil Engineering, Shanghai Jiao Tong University, Shanghai 200240, China

[2] Department of Civil and Environmental Engineering, College of Engineering, Shantou University, Shantou 515063, China

[3] Civil and Infrastructure Engineering Discipline, School of Engineering, Royal Melbourne Institute of Technology (RMIT), Victoria 3001, Australia

\* Correspondence: shensl@stu.edu.cn; Tel.: +86-754-8650-4551; Fax: +86-21-6419-1030

**Abstract:** Construction accidents are a significant hazard to the community, affecting sustainable development. This paper summarizes the safety situation of the construction industry in China over the past ten years. Detailed analysis is performed on fatal accidents that occurred in 2018 to reveal the spatiotemporal distribution pattern and characters of construction safety accidents. The construction failures are mainly attributed to management aspects rather than technical aspects. A case involving a major accident during shield tunnel construction in Foshan, Guangdong, in 2018 is investigated in detail. Strategic environmental assessment (SEA) is used to analyze the management issues of the Foshan metro project during planning, geological investigation, design, and implementation of construction works. The SEA result shows that the safety risk was very high with a low total SEA score. Based on the analysis, a guideline for safety construction management for sustainability is proposed.

**Keywords:** construction; accident; project management; strategic environmental assessment; China

---

## 1. Introduction

China has experienced rapid economic development over the last two decades, especially in urban infrastructure construction, which is a miracle of human achievement. Currently, the construction industry plays a crucial role in the national economy of China. The GDP output value from the construction industry was 23.5 trillion RMB¥ ($USD3.4 trillion) in 2018 [1], whereas China's total GDP was 90 trillion RMB¥ ($USD13 trillion) [2]. In 2018, the housing construction area increased to 14.09 billion square meters [1]. However, this rapid development came at the cost of the collapse of the environment and human health. Although China's construction industry has continued to develop in recent years, it is still labor-intensive and serves as a typical example of an extensive economy, due to its low technology use and unbalanced development. Some situations caused the collapse of ecosystems and even loss of life due to safety failures [3–5]. In recent years, not only natural hazards [6] but also construction-induced accidents have threatened safety and sustainable development [7,8]. Thus, this is not a sustainable development mode for the environment, the economy, and society. Sustainability means making sustained improvements to peoples' quality of life. Kelly [9] stated that "the construction and engineering sectors have a hugely important role to play in delivering the infrastructure for a sustainable future". The Institution of Civil Engineers [10] state that "sustainable development is now absolutely central to civil engineering and we must organize ourselves accordingly".

The construction industry in China features scattered construction sites, complex construction environments, a large number of personnel, and high mobility [11]. These features of China's construction industry result in high frequency of safety accidents during construction activities. However, the safety of construction engineering is of great significance for the sustainable development of the economy and social stability. The Chinese government identified these problems and tried to change this development mode recently. In recent years, the Chinese government issued and amended several Laws and regulations on environmental protection and sustainable development, e.g., Construction Law [12], Safety Production Law [13], Environmental Protection Law [14], and Urban and Rural Planning Law [15]. These laws and regulations tried to decelerate the construction speed and advocated strategic environmental assessment (SEA) during the planning, design, and implementation of construction works [16,17].

The construction industry is high-risk and can cause significant impacts to the environment and human health [18–20]. In response, with the rapid development of the construction industry, there have been many studies on various topics aiming to improve safety in the construction industry. The research topics can be mainly clustered into three major groups, namely, the safety management process, the impact of individual and group or organizational characteristics, and accident data [21]. From the perspective of the safety management process, research has become more comprehensive on safety measures [22,23], safety assessments [24,25], safety knowledge [26,27], safety monitoring [28,29], etc. The study on the characteristics of stakeholders' involvement in construction, namely, site workers [30,31] and groups or organizations [30,32], is crucial to enhance construction safety. Moreover, accident data is the basis of many studies, including accident statistics [20,33], accident cost analysis [34], and accident causation [35]. This paper belongs to the latter research group, though incorporates accident statistics and management analysis.

This paper summarizes the fatal accidents in the construction industry in China over the past ten years. The objectives of this study are: (i) to recognize the status quo and trends of safety in the construction industry in China; (ii) to analyze the features of accidents; (iii) to provide a basis for the government to establish the management regulations for sustainable development by reporting and analyzing the only major accident based on SEA; and (iv) to propose the mitigation measures for project management regarding accidents.

## 2. Background

Figure 1 summarizes the fatal accidents related to production safety in the construction industry in the past ten years in China. As shown in Figure 1, from 2009 to 2015, the numbers of accidents and fatalities showed a clear decreasing trend annually, which indicates that production safety gradually improved in general. However, the number of accidents per annum from 2016 to 2018 increased significantly. A reason for the sudden increase in accidents in 2016 is that the way accident data is reported was adjusted [36]. The validation date of the former reporting regulations for production safety accidents, established in 2014, became invalid in 2016. A more comprehensive statistical report regulation on production safety accidents was issued and implemented in 2016 [37]. The number of non-fatal, small accidents and number of deaths after an accident should be included in the report data under the new reporting regulations, which led to a significant increase in the number of reported accidents and deaths. However, the main reason for the annual increase in the numbers of accidents and fatalities since 2016 is the ineffective governance in the safety management. It is worth noting that the data in Figure 1 is from MOHURD [36], which does not fully cover all accidents for the problems in the accident reporting system and its implementation in China. Although some less serious accidents may not be reported, the data is still important for research on the overall trends.

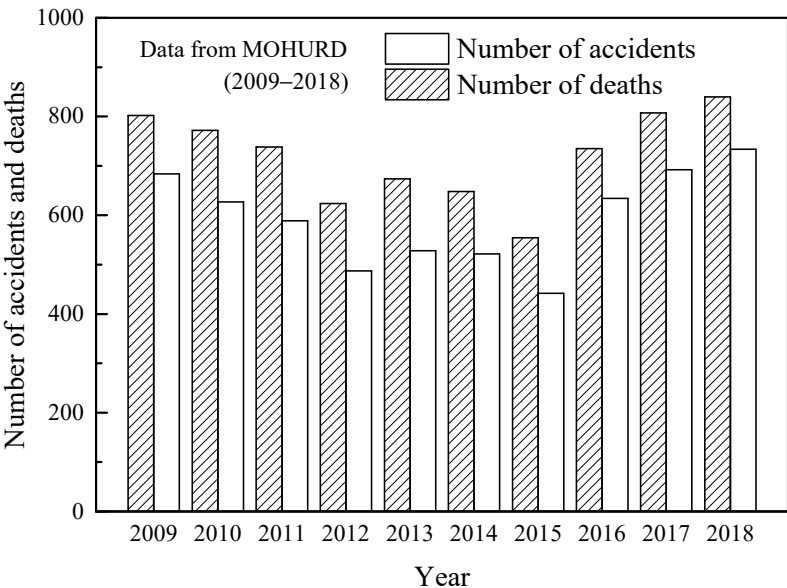

**Figure 1.** Fatal accidents related to production safety from 2009 to 2018 in China.

Table 1 shows the accident classification standards in China. The classifications are defined according to the following three aspects: death toll, seriously injured toll, and direct economic loss [38]. As summarized in Table 1, accidents are classified into four levels: particularly serious accidents, major accidents, large accidents, and general accidents. The data in Figure 1 includes four levels of accidents. According to Table 1, in 2018 there were 734 production accidents and 840 deaths in construction engineering projects, which include 1 major accident and 21 large accidents, with a total of 87 deaths. The number of large accidents decreased by two but the total number of accidents increased by 42 from those in 2017. There were no particularly serious accidents in China in 2018. Overall, within the past 10 years the situation of production safety in construction projects was generally stable in China, whereas the past 3 years has seen an increasing number of accidents and fatalities.

**Table 1.** Accident Classification Standard (data from GOSCPRC, 2007 [38]).

| Accident Level | Death Toll ($D$) | Seriously Injured Toll ($SI$) | Direct Economic Loss ($DEL$, in Million RMB¥) |
|---|---|---|---|
| Particularly serious accident | $30 \leq D$ | $100 \leq SI$ | 100 m $\leq DEL$ |
| Major accident | $10 \leq D < 30$ | $50 \leq SI < 100$ | 50 m $\leq DEL < 100$ m |
| Large accident | $3 \leq D < 10$ | $10 \leq SI < 50$ | 10 m $\leq DEL < 50$ m |
| General accident | $D < 3$ | $SI < 3$ | $DEL < 10$ m |

## 3. Analysis on Safety Accidents

### 3.1. Accident Types

According to MOHURD (2018), the production safety accidents related to construction engineering can be divided into the following six types: collapse accidents, falling from heights, struck by objects, mechanical injury accidents, crane-related accidents, and other types of accidents [39]. The "other" types include vehicle injuries, electric shocks, poisoning, fires, and explosions. Figure 2 shows the number and proportion of various types of accidents in China in 2018, including the four levels of accidents. As shown in Figure 2, the percentage of falling accidents is the largest, accounting for 52.18%, while being struck by objects and mechanical injury accidents reached 15.2% and 5.86%, respectively. The percentages of crane-related accidents and collapse accidents were similar (both around 7.4%), whereas the percentage of other types of accidents was 11.85%. Falling from heights was the most

frequent type of accident in the construction industry, accounting for more than half of the total number of accidents occurring in 2018.

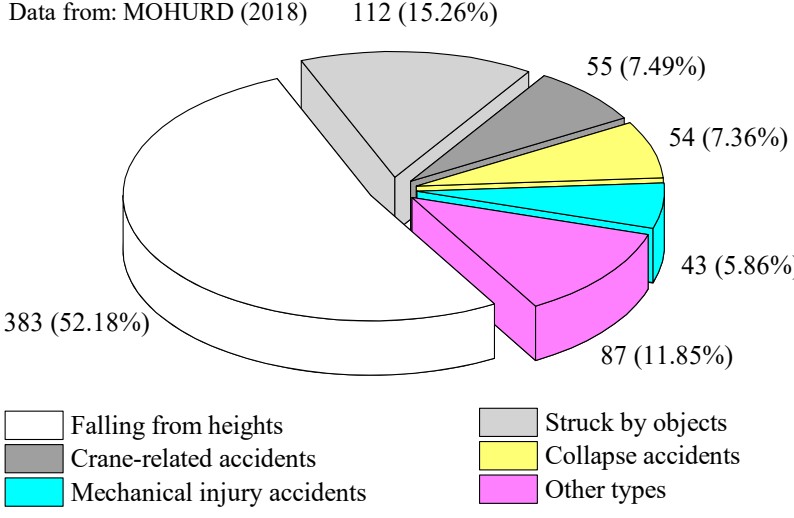

**Figure 2.** Accident types in 2018.

Figure 3 illustrates the number and proportion of large accidents in 2018. As shown in Figure 3, 21 large accidents in 2018 included 9 collapse accidents, 4 crane-related accidents, 2 falling from heights accidents, 1 case of mechanical injury accident, 5 cases of other types of accident, including 1 case of electric shock, 1 case of landslide, and 3 cases of poisoning and asphyxiation. Although there were 112 cases of people being struck by objects, no large accidents of this type occurred in 2018. The number of collapse accidents was ranked fourth among the total accidents, however, first in the category of large accidents, which implies that the consequences of collapse accidents are often more serious.

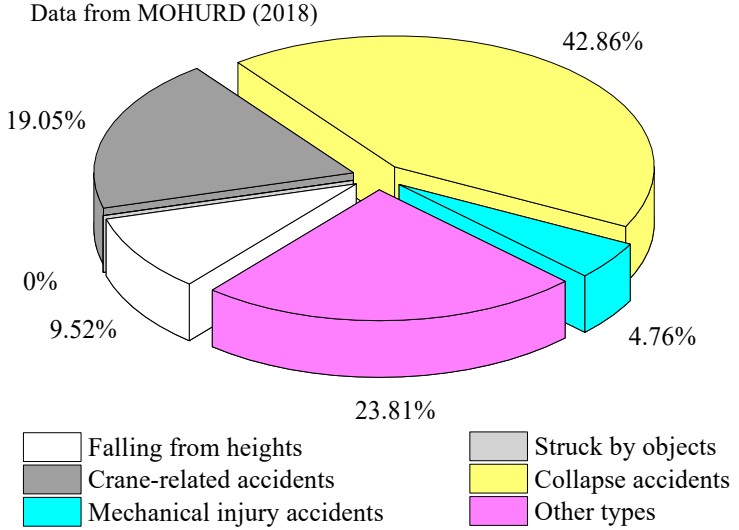

**Figure 3.** Proportion of large accidents in 2018.

### 3.2. Time Period of Accidents

Figure 4 demonstrates the monthly distribution of four levels of accidents with death numbers in 2018. As shown in Figure 4, the accidents in 2018 mainly occurred in April to September, whereas the number of accidents was lower after October and before March. The month with the lowest number of accidents was February. This can likely be attributed to the traditional Chinese Spring Festival in February, during which most construction projects were suspended.

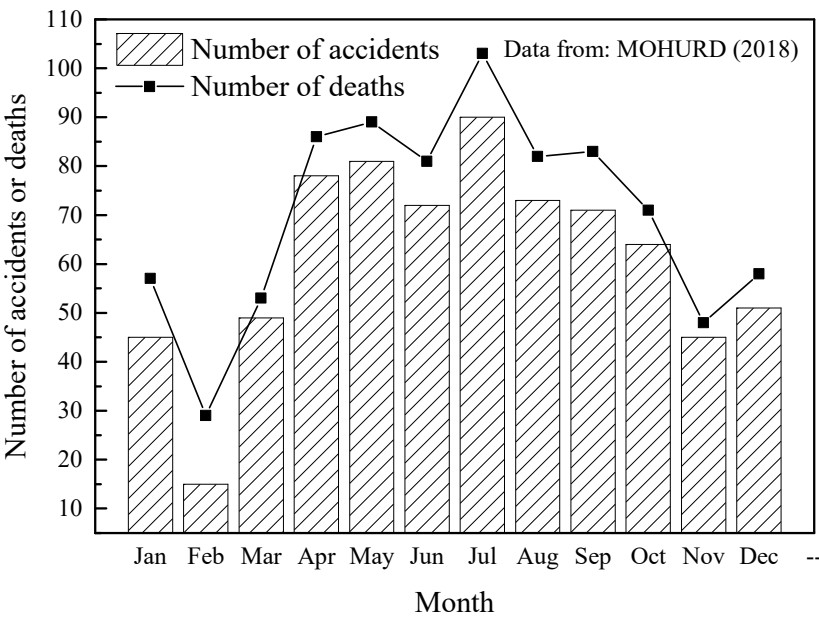

**Figure 4.** Monthly distribution of accidents with death number in 2018.

The time distribution for large accidents with deaths in 2018 is presented in Figure 5. As shown in Figure 5, the large accidents in 2018 mainly occurred in the morning. No accidents were reported around noon, because in China, midday is generally a rest time. There were still some accidents reported after 18:00, which may be due to overtime work during tight construction periods.

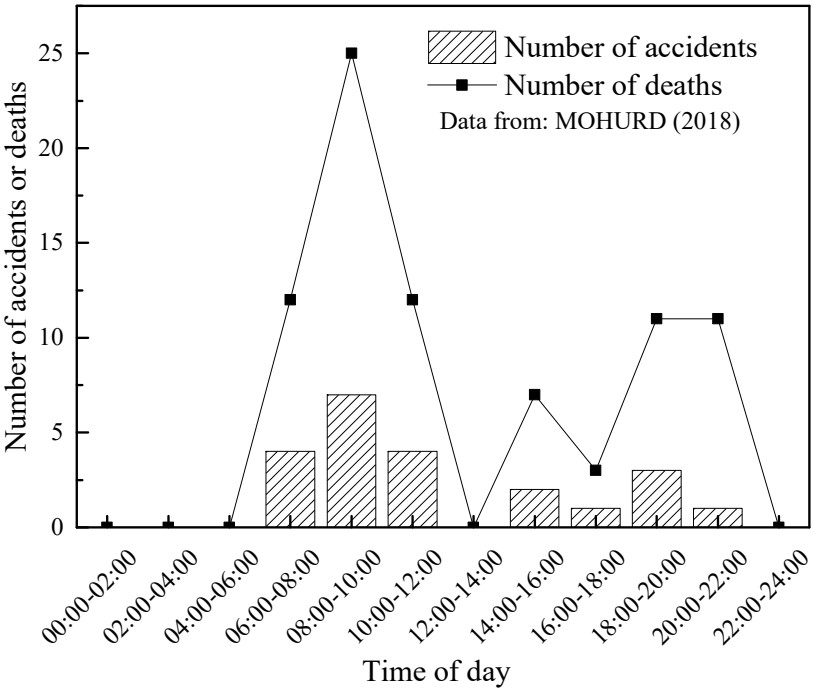

**Figure 5.** Daily distribution of large accidents in 2018.

### 3.3. Provincial Distribution of Accidents

The provincial distribution of production safety accidents was uneven in 2018. The provincial distribution was closely related to the economic development status and supervision intensity of each province. Figure 6 shows the number of accidents and gross domestic product (GDP) in different

provinces of mainland China from 2016 to 2018. The top ten provinces in terms of number of accidents were Jiangsu, Guangdong, Chongqing, Sichuan, Anhui, Zhejiang, Fujian, Heilongjiang, Gansu, and Hubei, according to the accident statistics in 2018. It is noted that four provinces (Sichuan, Heilongjiang, Fujian, and Gansu) were newly ranked in the top ten in 2018. Furthermore, the number of accidents in these four provinces increased annually from 2016 to 2018. This statistical result can be attributed to the fast-booming economy and inadequate security awareness in these provinces from 2016 to 2018.

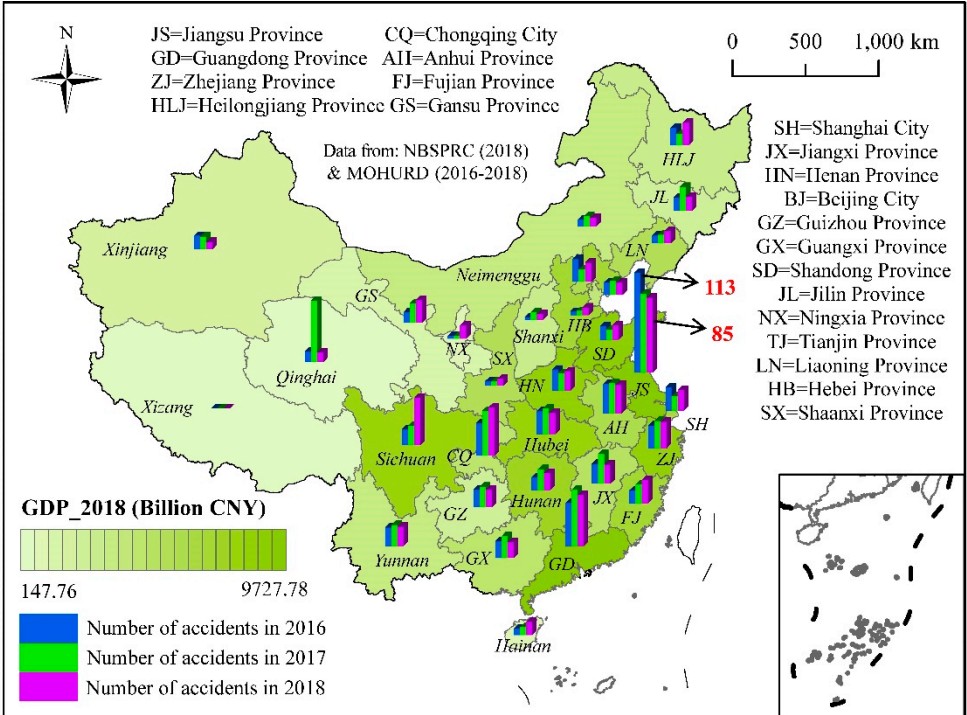

**Figure 6.** Number of accidents in 31 regions of China from 2016 to 2018.

Although the number of accidents in Jiangsu, the second province in terms of provincial GDP, was the highest in recent years, its safety situation has improved, as the trend shows a clear decrease. In addition, the number of accidents and deaths in 2018 in Ningxia, Hebei, Hainan, Shaanxi, Beijing, Qinghai, Shanghai, Shandong, and Liaoning increased compared to in 2017. Most of these provinces, such as Beijing, Shanghai, and Shandong, feature high GDP outputs. Generally, the economically developed provinces are at greater risk of production safety accidents due to the large number and large scale of construction projects. Some areas with relatively low economic development, such as Hainan, Ningxia, and Qinghai, experienced rapid development due to the increase in government investment in recent years, and therefore, the construction industry in these provinces expanded accordingly. However, the technology and management of production activities in these provinces did not develop simultaneously. As a result, the number of production safety accidents also increased.

In 2018, fifteen provinces in China suffered from large or major safety accidents in the construction industry. One major accident and two large accidents occurred in Guangdong Province, with a total of 19 deaths. Although there was only one major accident in China in 2018, it still caused great loss to life and property—this was the Foshan shield tunnel collapse accident. The details of this major accident are reported in the following chapter to explain the background, development process, and causes of this accident.

## 4. A Case Study of Major Accident

On February 7, 2018, a collapse accident occurred (8:40 p.m.) during the shield tunnel construction of Huyong Station to Ludaohu Station in the first phase of rail transit line 2 in Foshan City, Guangdong Province. Figure 7 illustrates the geographical location of this safety accident, showing the ground collapse. The ground collapse was approximately 6 m to 8 m in depth. The area of the ground collapse was about 4192 $m^2$ and the volume of the collapse was close to 25,000 $m^3$ [40]. Based on the standards shown in Table 1, this accident was classified as a major accident, which resulted in 11 deaths, 1 missing person, 8 injuries, and about 53.238 million RMB¥ ($USD7.724 million) direct economic loss. Yu et al. [41] investigated the technical issues related to the accident. The results showed that this accident was caused by the leakage of the brush seals at the tail of the shield machine [41]. This collapse accident was the only major accident in the construction industry in 2018. The post-accident report showed that there were many management problems in this project [40].

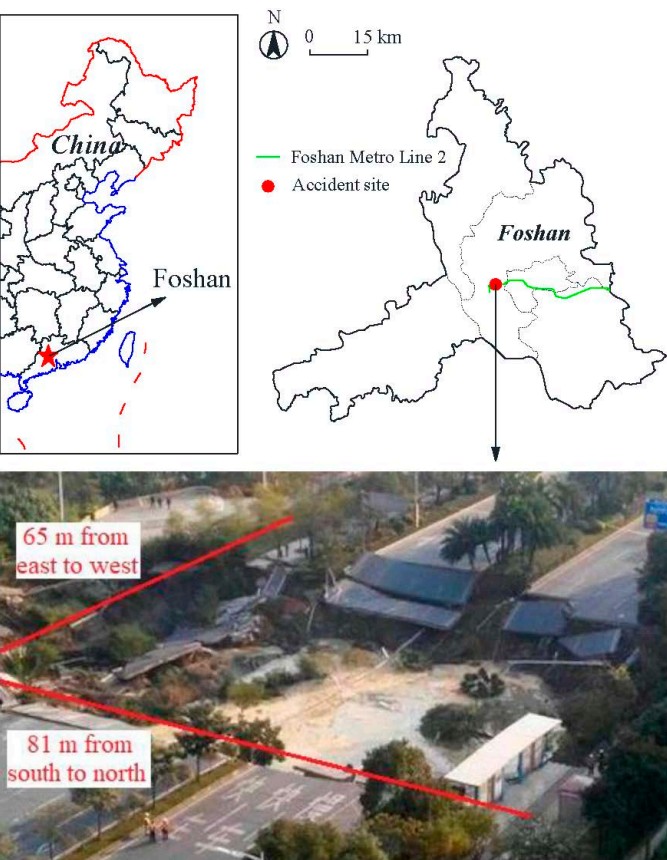

**Figure 7.** The collapse during tunnelling in Foshan metro line 2 in 2018, induced by groundwater leakage.

### 4.1. Accident Process

Table 2 tabulates the development process of the accident over time. On 7 February 2018, the shield machine was located in a complex geological environment at a depth of about 30.5 m, where the ground contains mucky silt, silty sand, and medium sand, with confined water. During the assembly operation of the segment, the soil warehouse pressure suddenly increased and the tail of the shield machine appeared to sink with the seepage. Although the construction workers immediately took urgent plugging measures, the leakage and sand blasting area continued to expand, which caused the shield machine and segment structure to deform downward. Once the tunnel structure failed, a huge amount of the muddy sand suddenly poured into the tunnel, which pushed the shield machine trolley back approximately 700 m. Meanwhile, the mud-sand flow and the associated air waves knocked down or buried some of the workers, which caused heavy casualties.

**Table 2.** Development of the collapse accident in Foshan metro line 2 on February 7, 2018 (data from EMBF, 2018 [40]).

| Time Line | Critical Events |
|:---:|:---|
| 18:10 | Excavation of the 905th ring was finished and shield tail was cleaned. |
| 18:52 | The first piece of the segment was assembled. The pressure of the soil chamber increased from 233 kPa to 276 kPa. The tail of the shield machine began sinking. The right side of the first segment burst upward. |
| 18:54 | The slurry flowed over the surface of the remaining segments. |
| 19:03 | The situation was reported and emergency measures were taken. |
| 19:47 | Personnel failed to control the danger. |
| 20:03 | The vertical deviation of shield tail reached −460 mm. |
| 20:35 | Personnel in the tunnel began to evacuate. |
| 20:36 | The muddy sand flow sprayed around the 899th ring segment. The shield tail sank 463.5 mm. |
| 20:40 | A large area of ground collapsed. A huge amount of muddy sand poured into the tunnel. The mud-sand flow and air waves hurt some workers trying to escape. |

*4.2. Accident Causes*

According to the investigation from the Guangdong Provincial Government [40], there are three direct technical causes of the accident [41]: (i) the geological and hydrological conditions at the site were very challenging, and therefore the risk of groundwater leakage was high when the shield machine passed through this section; (ii) the sealing device at the tail of the shield machine did not work well during the construction process. Some seepage channels were generated under high external water and earth pressures; and (iii) after the tunnel structure failed, a large amount of muddy sand swiftly entered the tunnel, forming strong mud-sand flows and air waves along the longitudinal direction of the tunnel.

In addition, the government investigation showed the management causes of the accident [40]. This was an engineering procurement construction (EPC) project. Figure 8 shows the EPC management relationship chart. For comparison, the original mode of EPC project management is also presented in Figure 8a. As shown in Figure 8b, there were 12 parties involved in the construction management of this project, involving investment, construction, supervision, operation, exploration, design, and labor dispatch. As shown in Figure 8, compared to the original EPC mode, there are many management levels, resulting in increasing safety and environmental risks. For example, the right-line shield machine experienced several instances of shield tail leakage before the accident, which presented great risk to the shield tunneling. The construction party did not eliminate this safety hazard. As a violation of the Labor Law [42], the working hours of construction site staff were extended in this project. Moreover, safety production inspections were not carried out regularly. There was inadequate supervision of some government departments, such as the transportation bureau, fire station, and production safety supervision bureau. The project was a safety production risk point in Foshan city, nevertheless it was constructed without the relevant formalities, such as construction project planning permit, and fire design plan reviews. Even worse, some construction workers continued to carry out rescue work in the tunnel in this dangerous situation, which led to serious casualties. Technical problems were the direct cause of this collapse accident, whereas adequate safety management could have avoided the accident.

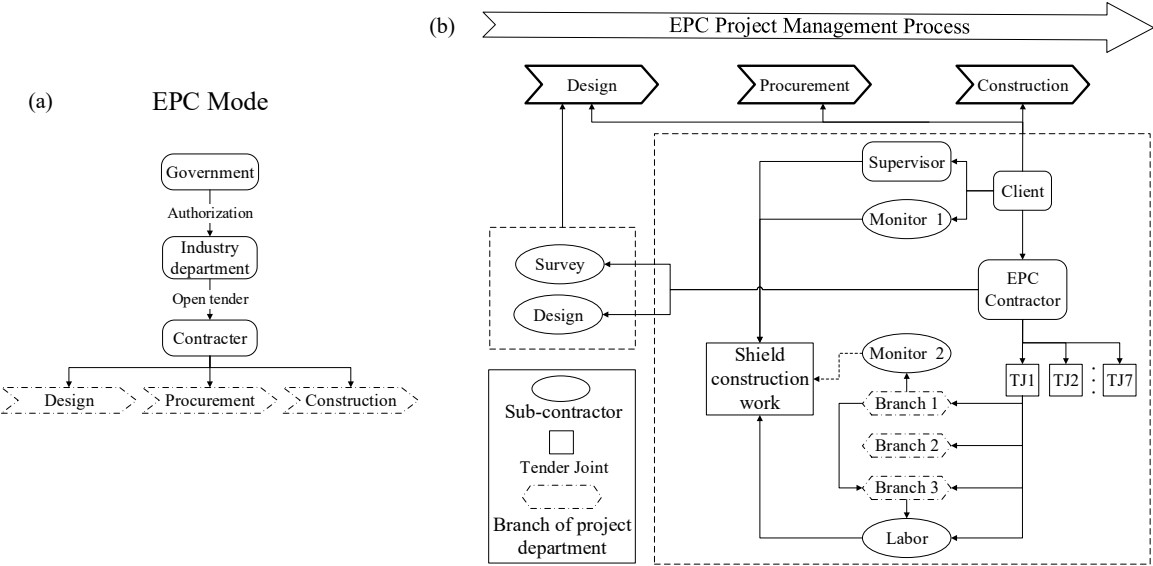

**Figure 8.** The EPC project management relationship chart (data from EMBF, 2018 [40]). (**a**) the original mode of EPC project management, (**b**) the EPC management situation of Foshan Metro Line 2.

### 4.3. SEA Analysis

Strategic environmental assessment (SEA) was proposed and applied to analyze sustainable development problems related to the social, economic, cultural, human health, and environmental [43] factors at a strategic level. Shepherd and Ortolano [44] summarized six principles for SEA effectiveness in promoting sustainable urban development. To check the risks of metro tunnel construction at the strategic level, SEA analysis was used in this paper and applied to the construction management of Foshan metro line 2.

The evaluation of the project management of Foshan Line 2 is presented in Table 3. The score is given according to the following management factors: (1) sustainability principles were considered in municipal management and urbanization phases, but were insufficient during the construction phase; (2) there was no simultaneous assessment of the impact on the surrounding environment after the construction started; (3) many factors were not considered relating to the cumulative impacts; (4) there was a problem with sealing of the shield machine that was not effectively solved; (5) land subsidence monitoring was carried out, with 19 alarms issued from 1 June 2017, to 7 February 2018, but inappropriate disposal methods were taken when dangerous situations occurred; (6) although there were a lot of government and company regulations, they are not well implemented during the construction process, and the supervision work was inadequate. As shown in Table 3, only 12 out of a total of 30 points are obtained, indicating that the environmental and construction risk of Foshan Line 2 was very high based on the SEA analysis.

**Table 3.** Specific evaluation of Foshan line 2 using SEA principles [44].

| SEA Principle | Project Management of Foshan Line 2 | Score Out of 5 |
|---|---|---|
| 1 | Yes: Sustainability principles were generally considered. | 3 |
| 2 | Not perfect: Most of the assessment was conducted just before the project. | 2 |
| 3 | Not perfect: Multiple and correlated impacts were not considered comprehensively. | 2 |
| 4 | Not perfect: Did not adhere to the principle of sustainability during the construction process. | 2 |
| 5 | Not perfect: The ground settlement was monitored during construction. However, there was no appropriate feedback or disposal measures. | 2 |
| 6 | Not perfect: Legal and public monitoring mechanisms were not implemented | 1 |
| Total score | | 12 |

Note: If SEA principles are fully and positively followed, a full score of 5 is awarded for each principle; the total score for the SEA principles is out of 30.

### 5. Discussion

To ensure sustainable development, it is crucial to analyze previous accidents. This collapse was judged a liability accident by the Guangdong provincial government [40]. This accident revealed some potential project management weaknesses. With a low total score of only 12 out of 30 for the SEA assessment, the safety risk of this project was very high. The contractor showed insufficient management of safety risks, improper emergency response, and untimely evacuation of personnel, which were all responsible for this accident. In addition, effective technical and management measures were not taken to eliminate hidden risks, e.g., the defective performance of shield tail seals. To avoid similar accidents in the future, the following guidelines for risk management based on the conducted analysis are proposed (see Figure 9).

(i)　Step I: Incorporation of SEA. We propose the SEA concept be incorporated into project management. Strategic risk assessment should be conducted at the project level. The total score of the SEA should be higher than 24. He et al. [14] verified that when an SEA score increased from 16 to 24, the environmental risk (land subsidence) was reduced greatly.

(ii)　Step II: Limitation of Construction Speed. Based on SEA results, the appropriate construction speed of the project should be discussed and determined by the supervising government body, the client, and the contractor.

(iii)　Step III: Strengthen Safety Management. There are many subcontractors with loose connections, and it is crucial to reduce the numbers of subcontractors. The training of the laborers should be improved. However, when the number of subcontractors cannot be reduced, supervision and monitoring should be improved during implementation. An operational manual should be distributed to all engineers and workers in the field. Moreover, overtime work in the field should be limited.

(iv)　Step IV: Establishment of Monitoring and Early Warning System. The full monitoring system should include not only field monitoring but should also establish a working group application for cell phones. An overview of the field situation should be uploaded each day, and when a risk warning is received, this information should be uploaded every minute. A comprehensive safety early warning system should be established to replace the passive after-the-fact safety management model with active advanced warning.

After these guidelines are implemented, the risks will be reduced greatly.

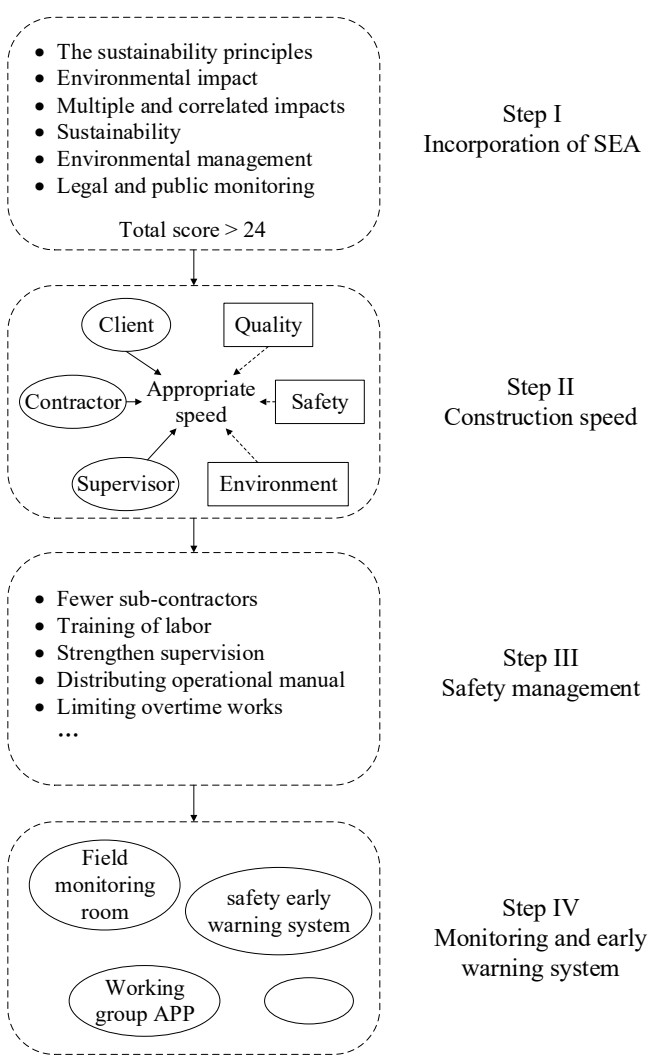

**Figure 9.** The guidelines for risk management.

## 6. Conclusions

This paper analyzed the production safety in the Chinese construction industry. The management reasons were investigated in detail via the SEA concept. Guidelines for risk-control management in construction were proposed to ensure community sustainability. Major conclusions are drawn as follows:

(1) Safety accidents cause great losses of life and property, which expose the problems in construction management hindering the sustainable development of society. Over the past ten years, the number of fatal accidents first steadily declined from 2009 to 2015, however the production safety situation during the most recent three years became worse. In addition, large accidents that cause mass casualties have not been completely prevented in China.

(2) The analysis of the accidents in 2018 shows that safety development was distributed unevenly in China. Jiangsu province had the largest number of fatal accidents in recent years, but the situation improved in 2018. The seasonal distribution of fatal accidents shows that the lowest number of accidents occurred in February, and the accidents mainly occurred in the morning over a one-day period. The most frequent accident type was falling from heights; nevertheless, the most common large accidents were collapses.

(3) The management reasons were investigated through a field failure case study of the shield tunnel construction in Foshan, Guangdong Province. SEA was used in an environmental impact

assessment to analyze the safety risks for this case. According to the SEA results, inadequate management of this project led to high safety risks. Inadequate safety management often leads to the deterioration of other factors, e.g., personnel, facilities, and environmental factors, and ultimately lead to accidents. At the same time, proper safety management can identify and eliminate various potential hazards. Therefore, it is important to enhance safety management during construction.

(4) To ensure production safety in the future, a guideline based on SEA was proposed, including the following 4 aspects: (i) an SEA score over 24; (ii) limitation of construction speed; (iii) strengthened safety management.; (iv) establishment of monitoring and early warning systems. Moreover, the accident reporting system is crucial in studying accidents, which is closely related to the experience of the accidents. Thus, an integral accident reporting system, including fatal, trivial, and even potential accidents, detailed classification, and comprehensive impacts, is also highly recommended.

**Author Contributions:** This paper represents a result of collaborative teamwork. S.-L.S. developed the concept of the manuscript. X.-H.Z. drafted the manuscript. Y.-S.X. provided constructive suggestions and revised the manuscript. A.-N.Z. revised the manuscript. The four authors contributed equally to this work.

**Funding:** The research was funded by the Innovative Research Funding of the Science and Technology Commission of Shanghai Municipality (Grant No. 18DZ1201102), and the National Natural Science Foundation of China (NSFC) (973 Program: 2015CB057802).

**Acknowledgments:** The authors would like to express their sincere appreciation to the anonymous reviewers and editors, whose constructive suggestions helped the authors to rethink and rework the manuscript to improve the quality greatly.

**Conflicts of Interest:** The authors declare no conflict of interest.

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
