# Peer review of "Analysis of Production Safety in the Construction Industry of China in 2018"

_sustainability, doi:10.3390/su11174537_

Round 1
Reviewer 1 Report
Manuscript: "Analysis of production safety in construction industry of China in 2018"
This work presents an overview of the construction work safety in China, and suggests some approaches for improvements. I suggest this manuscript to be accepted for publication after minor revision. The manuscript is well written and easy to follow. However, it leaves me with some questions: the authors should consider the following comments:
Abstract: You write "Based on the analysis, several safety recommendations for sustainable construction are proposed". Could you maybe be clearer on what type of recommendations you propose? Now it sounds like you are proposing even technical solutions, but you are not doing anything near that in the work. In the Discussion (Line 188) you state those recommendations and they can be summed up by "government and companies should pay more attention to production safety"? It's quite vague and not very detailed recommendations, but in the Abstract you write that several safety recommendations were proposed?
Line 47: You mention here that the ways accidents are reported have been changed, and that answers to why the accident/death toll increased from 2016. Could you say more about how these reporting changes affects this increase seen in Figure 1?
Figure 1- 4: In table 1 you define how accidents are characterized: general - large - major - particularly serious. Are figure 1, 2 and 4 showing data for all classifications of accidents, summed up? And Figure 3 say it's for 'large' accidents: is it only large, not large and "up" (major, particularly serious included)? You could be clearer on what these Figures show. Also, are really the death toll higher than amount of reported accidents, or are you only presenting data for accidents leading to fatalities?
Line 188: Here you state the conclusions from this study "...this study proposes three recommendations based on the analysis..." but this should also be stated in the conclusion and perhaps in the abstract too.
Overall comment: Just a reflection from me: Could a suggestion also be to implement an accident reporting system that ensure that all accidents and serious incidents are reported, including events not leading to any injuries but could have led to injuries, and thereby get more data to work on for all stakeholders? First, the reason for good work safety is to protect the people working there. Secondly, it is very financially lucrative for government and companies to make sure that the work place is safe. If a working platform falls down it should be reported even if no one is injured, so that it can be used to continuously improve the work safety and prevent future costly accidents. Even smaller incidents (tool/equipment failures/mishandling) that lead to injuries or any disruption in the process should be reported. For example, I just searched and found a selection of data for a Scandinavian country, where all reported accidents in the construction industry for specific year (2014) was shown: 5823 accidents reported. The death toll was 6 workers. It is also showed how many of the accidents led to short sick-leave, long-sick leave, invalidity. The data also is separated into different groups of workers at construction sites: general builders, metal-, wood-, concrete workers, electricians, painters etc. In the data you show from China, it appears as if the only accidents reported are the bigger ones that lead to fatalities: The death-toll is higher than number of accidents. I would say that a first, fundamental suggestion to how to improve work safety would be to suggest start reporting and continuously work on the knowledge attained from all incidents that have/or could have caused serious accidents.
Reviewer 2 Report
The topic is of relevance and interest but needs to be framed so that it is of general relevance to other regions and countries.
Currently the abstract summarises the paper briefly. It needs to detail the aim, method and the significance of the study.
There needs to be a critical literature review to understand how the safety regulations are in China as compared to other countries. There is no literature review or theoretical framework to locate this study in literature
The reason for using a case study approach in this paper needs to be explained. The MOHURD data appears to be the primary source of data. Should there be more sources?
There needs to a more explanation of the method. Why was the Huyong Station case selected; how was the data collected; what was the analysis based on? The source of information for the data is not clear. Is it being based only on the government records or any other sources were used too?
The case study does not add much to the conclusions. The conclusions are resulting more from the Background section and the Analysis section.
A discussion of what the current policies are regarding safety also need to be included to understand whether the polices are inadequate or the it was an implementation failure.
